# Learning from COVID-19: Infectious Disease Vulnerability Promotes Pro-Environmental Behaviors

**DOI:** 10.3390/ijerph18168687

**Published:** 2021-08-17

**Authors:** Da Jiang, Mingxuan Li, Hanyang Wu, Shuang Liu

**Affiliations:** 1Department of Special Education and Counselling, The Education University of Hong Kong, Tai Po, New Territories, Hong Kong, China; lshuang@eduhk.hk; 2Integrated Centre for Wellbeing, The Education University of Hong Kong, Tai Po, New Territories, Hong Kong, China; 3Centre for Psychosocial Health, The Education University of Hong Kong, Tai Po, New Territories, Hong Kong, China; 4Department of Business Administration, School of Management, Shenzhen University, Shenzhen 518060, China; lmx000430@outlook.com; 5Department of Finance, School of Economics and Management, Jilin Institute of Chemical Technology, Jilin 132000, China; Why851223@163.com

**Keywords:** pro-environmental behavior, infectious disease vulnerability, empathy

## Abstract

Environmental problems, such as climate change, pollution, and environmental degradation, are important contributors to the spread of infectious diseases, such as COVID-19 and SARS. For instance, a greater concentration of ambient NO_2_ was associated with faster transmission of the SARS-CoV-2 virus, which causes COVID-19. However, it remains unclear whether outbreaks of infectious diseases arouse individuals’ concern on the need to protect the environment and therefore promote more pro-environmental behaviors. To this end, we examined the relationship between infectious disease vulnerability and pro-environmental behaviors using data from a cross-societal survey (*N* = 53 societies) and an experiment (*N* = 214 individuals). At both the societal and the individual levels, infectious disease vulnerability increased pro-environmental behaviors. At the societal level, this relationship was mediated by citizens’ level of environmental concern. At the individual level, the relationship was mediated by empathy. The findings show that infectious disease vulnerability is conducive to pro-environmental behaviors.

## 1. Introduction

Coronavirus disease 2019 (COVID-19) has had an unprecedented global impact. Its rapid spread has harmed the physical and psychological health of billions of individuals worldwide [1,2,3]. As of 10 August 2021, more than 203 million people worldwide had been confirmed to be infected. Evidence from the spread of other serious infectious diseases, such as severe acute respiratory syndrome (SARS) and Middle East respiratory syndrome, suggests that environmental problems, such as climate change, pollution, and environmental degradation, are important contributors to the spread of infectious diseases. However, do outbreaks of infectious diseases in turn arouse individuals’ awareness of the need to protect the environment? Although a link between environmental problems and the spread of infectious diseases has been established [4,5,6], it remains unclear whether infectious disease vulnerability increases pro-environmental behaviors. The current study aims to fill this gap by examining the relationship between infectious disease vulnerability and pro-environmental behaviors using data from a cross-societal survey and an experiment.

### 1.1. The Environment and Infectious Diseases

Environmental factors are important reasons for the spread of infectious diseases. Based on 20 experiments with guinea pigs, Lowen et al. found that both ambient relative humidity and temperature influence the aerosol spread of influenza virus [7]. Tan et al. (2005) found that a sharp rise or fall in environmental temperature significantly increased the number of identified SARS cases, because the human immune system may be weakened by sudden changes in temperature [8]. Cui et al. (2003) conducted ecological analysis in five regions of China and found that the fatality rate for SARS was positively associated with an increase in the air pollution index (where higher values indicate a more severely polluted environment) [4]. Kan et al. (2005) found that increased moving averages of three air pollutants (particulate matter 10 micrometers or fewer in diameter (PM10), sulfur dioxide, and nitrogen dioxide (NO_2_)) were associated with a higher risk of daily SARS mortality from 25 April to 31 May 2003, the incubation period following the first reported case of SARS in Beijing [5]. In a recent study, Yao et al. found that a greater concentration of ambient NO_2_ was associated with faster transmission of the SARS-CoV-2 virus, which causes COVID-19, in cities both inside and outside Hubei province [9]. In addition, city-level PM2.5 was associated with a higher COVID-19 fatality rate in cities both in and outside Hubei province [9]. These studies showed that environmental factors are significantly associated with the spread of infectious diseases.

However, do infectious diseases arouse individuals’ awareness of the importance of protecting the environment and further promote more pro-environmental behaviors? Recently, Reese and colleagues applied the social identity model of pro-environmental action (SIMPEA) to the context of COVID-19 [10,11]. The SIMPEA argues that individuals’ identification towards social groups predicts their appraisal of a crisis. Their appraisal of a crisis, in turn, results in changes to emotions and motivations toward pro-environmental actions. Reese [11] argued that COVID-19 as a crisis highlights individuals’ social identity (e.g., a family/group member) and their place identity (as a citizen of a country or a city), because of its risks and the measures related to controlling the virus. Closing public places may increase individuals’ preference for natural environments and therefore increases connectedness to natural environment. Because of such perceived connectedness, individuals may be more likely to conduct pro-environmental behaviors [12]. Therefore, Reese et al. argued that COVID-19 would increase pro-environmental behaviors. This argument, however, has not been empirically tested yet.

In addition, in the field of social psychology, Katz (1960) identified a utilitarian function as one of the important pathways to changing attitude and behaviors, based on the behaviorist theory of learning [13]. The concept of a utilitarian function of attitude suggests that seeking positive reinforcement (such as rewards) and reducing negative reinforcement (such as punishment) are key motivations for attitude change and behaviors. Individuals change their attitudes and behaviors when they perceive such attitudes and behaviors to have negative consequences [14]. In the context of disease vulnerability and environmental protection, individuals may learn from social media reports that environmental destruction causes outbreaks of infectious diseases. They may then consider epidemics a negative consequence of environmental destruction. To avoid this negative consequence, they may be more likely to change their behaviors to protect the environment. Indeed, Prokop and Kubiatko (2014) found that schoolchildren who perceived themselves to be more vulnerable to diseases reported stronger pro-environmental attitudes than did those who perceived themselves to be less vulnerable [15]. Prokop and Kubiatko (2014) explained this association by arguing that the behavioral immune systems of more vulnerable individuals are more sensitive to environmental threats or changes [12]. Therefore, such individuals feel compelled to invest more effort in protecting the environment to reduce their risk of being exposed to environmental threats. However, this study was based on a correlational survey of children. To the best of our knowledge, no study has examined the relationship between disease vulnerability and environmental attitudes with an adult sample or using an experimental design. The studies reported here aimed to fill these gaps.

### 1.2. The Role of Empathy

We hypothesized that perceived disease vulnerability increases pro-environmental behavior through the affective state of empathy. “Empathy” refers to the capacity to sense, understand, and even share another person’s feelings [16]. Viewing others’ vulnerability to disease and disasters significantly induces empathy for those others [17,18]. For instance, physicians, nurses, and medical students have reported feeling empathy with their patients [19,20,21]. Media portrayals of natural disasters have been found to generate empathy, encouraging people to provide prosocial support for those affected [17,22].

Batson (2014) proposed that inducing empathy can increase prosocial and altruistic attitudes and behaviors toward different objects and subjects [23]. Indeed, empathy has been shown to be an affective emotional approach to generate greater concern and understanding [16,24] toward a wide range of targets, including ethnic or racial minorities [25], homeless individuals [26], and individuals with AIDS [27], and dementia [28]. In addition, empathy has been found to induce prosocial behaviors in both rats [24] and human beings [29]. Rameson, Morelli, and Lieberman (2012) found that empathy was associated with greater activity in the medial prefrontal cortex, which was associated with more prosocial behaviors [30]. In particular, Schultz (2002) found that participants who were asked to adopt the perspective of an animal reported more biosphere-oriented environmental concerns than did participants in the control condition [31]. Using an experimental design, Berenguer (2007) found that inducing empathy increased pro-environmental attitudes and behaviors. Participants in the high-empathy condition were asked to imagine themselves as a tree or as a dead bird on a beach covered in oil, while participants in the low-empathy condition were asked to take an objective perspective [32]. The high-empathy participants allocated more funds to an association that worked to protect the environment than did the low-empathy participants. Taken together, these findings suggest that perceived infectious disease vulnerability may prime individuals’ empathy and therefore increase their engagement in behaviors to protect the environment. The current studies aimed to examine these possible relationships.

### 1.3. The Current Studies

We conducted two studies to examine the relationship between infectious disease vulnerability and pro-environmental behaviors. In Study 1, we analyzed societal-level data on pro-environmental behavior from the fifth wave of the World Value Survey (WVS) [33]. We then examined the association between the societal-level Infectious Disease Vulnerability Index (IDVI) and pro-environmental behavior. The IDVI was created by Moore et al. with seven domains: healthcare, public health, economic factors, disease dynamics, demographic factors, political–domestic factors, and political–international factors [34]. It describes the capacity of a society to manage infectious diseases by limiting their spread. A higher score in the IDVI indicates a greater capacity to respond to a pandemic. The IDVI has been used in a number of recent studies examining societal-level capability in handling the COVID-19 pandemic [3,35,36,37]. We hypothesized that the more vulnerable a society is to infectious diseases, the greater the pro-environmental concerns individuals in that society will express, and the more actively they will thus engage in pro-environmental activities.

Study 2 was an experiment in which we examined the causal relationship between infectious disease vulnerability and pro-environmental behavior at the individual level. In addition, we examined the potential role of empathy in mediating this relationship. In Study 2, we primed for perceived vulnerability to infectious diseases of participants in the experimental condition by showing the participants a set of video footage showing street view and people’s daily life during the COVID-19 lockdown in Wuhan (the city where the first confirmed case of COVID-19 was reported), China. Participants in the control condition viewed street view and individual life video footage of Hangzhou, China, recorded before 2020. Next, we examined individual participants’ willingness to engage in pro-environmental behaviors. We hypothesized that the participants in the experimental condition would be more willing to engage in pro-environmental behaviors than would the participants in the control condition. Based on previous findings regarding the relationship between disease vulnerability and empathy [38], and that between empathy and pro-environmental behaviors [32], we hypothesized that empathy would mediate the relationship between infectious disease vulnerability and pro-environmental behaviors, such that greater vulnerability would induce greater empathy and therefore encourage more pro-environmental behaviors.

## 2. Study 1 Method

### 2.1. Sample

We adopted data from the fifth wave of the WVS (2018) [33]. This wave of the WVS was administered from 2005 to 2008, with 80,054 respondents from 56 societies. The age range was very large in this sample, from 15 years to 98 years (*M* = 41.40 years, *SD* = 16.56 years). The percentages of male (49%) and female (51%) respondents were similar (39,203 males, 40,754 females, and 97 non-respondents). Across the regions and countries surveyed, the proportion of female participants ranged from 43% to 56%. See Table 1 for a list of the 56 countries or regions and detailed information on the key variables for each society. We used societal-level scores as independent and dependent measures in this study.

### 2.2. Instrument

#### 2.2.1. Pro-Environmental Behaviors

Wave 5 of the WVS asked the respondents to report their membership of a list of organizations. This item was adapted from a previous scale measuring membership of voluntary associations [39]. We used the WVS item measuring respondents’ degree of activity as members of environmental organizations to indicate their pro-environmental behaviors. The participants in our study were asked to indicate the degree of their activity as members of environmental organizations by choosing one of three options: 0, *not a member*, 1, *inactive member*, or 2, *active member*. The data of respondents’ pro-environmental behavior in India, Indonesia, and Mali were excluded in data analysis, because their scores were out values in boxplot in SPSS with 5% trimmed mean.

#### 2.2.2. Environmental Concern

The participants’ level of environmental concern was measured in terms of their willingness to sacrifice for environmental protection using three items (i.e., “I would give part of my income to protect the environment” (reversed item), “I would accept an increase in taxes if the extra money was used to protect the environment” (reversed item), and “The Government should reduce environmental pollution, but it should not cost me any money”). The respondents were asked to indicate the extent to which they agreed with each statement on a 4-point scale, from 1, *strongly agree*, to 4, *strongly disagree*. The responses given for these three items were aggregated. A higher score indicated a greater level of environmental concern. The value of Cronbach’s alpha for the three items ranged from 0.30 to 0.80 (*M* = 0.60, *SD* = 0.11) across the 56 societies.

#### 2.2.3. Infectious Disease Vulnerability Index

We adopted the IDVI scores for the 56 countries or regions from Moore et al. [35]. The IDVI is scaled from 0, *worst*, to 1, *best*, according to a society’s capacity to prevent and manage infectious disease threats. The IDVI scores for these 56 societies range from 0.18 to 1.00 (*M* = 0.70, *SD* = 0.19)

#### 2.2.4. National Wealth

Gross domestic product per capita in 2005 (GDPPC; World Bank, Washington, DC, USA, 2005) [40] was controlled for in the data analysis.

## 3. Study 1 Results

Controlling for GDPPC, there was a negative association between IDVI score and pro-environmental activity (*b* = −0.32, *SE* = 0.11, *p* = 0.007). Using the PROCESS Macro version 3.5 [41], we further examined the possible mediation of this effect. Consistent with our prediction, the results showed that the association between the IDVI and pro-environmental behavior was mediated by environmental concern (Figure 1). When environmental concern was added to the model, the IDVI (*b* = −0.23, *SE* = 0.11, *p* = 0.043, 95% CI = (−0.45, −0.01)) was significantly negative associated with pro-environmental behaviors, whereas environmental concern (*b* = 0.17, *SE* = 0.05, *p* = 0.002, 95% CI = (0.06, 0.28)) was significantly positively associated with pro-environmental behaviors. The IDVI was positively associated with environmental concern (*b* = −0.54, *SE* = 0.31, *p* = 0.086, 95% CI = (−1.15, 0.08)). The direct effect size of the mediation was −0.23 (95% CI = (−0.45, −0.01)). The indirect effect of IDVI on pro-environmental behaviors through environmental concern was −0.09 (95% CI = (−0.22, −0.002)). People from societies with a lower IDVI score (indicating greater vulnerability to infectious diseases) showed a greater level of environmental concern, and were therefore more active in environmental organizations (Figure 2).

Figure 1 Environmental concern as mediator of the relationship between Infectious Disease Vulnerability Index and pro-environmental behaviors in Study 1.

## 4. Study 1 Discussion

Analyzing a scalable data set of 53 societies, Study 1 provided evidence that when a society is more vulnerable to infectious diseases, the more active individuals in that society are as members of environmental organizations. This association was found to be mediated by a greater level of environmental concern, measured by willingness to allocate one’s own and the government’s money to address environmental problems. Study 1 was a correlational study that provided societal-level evidence. To examine the causal relationship between infectious disease vulnerability and pro-environmental behaviors at the individual level, we conducted Study 2, in which infectious disease vulnerability was experimentally manipulated.

## 5. Study 2 Method

### 5.1. Participant

Two hundred and fourteen participants (68% female; age range: 18–81 years, *M* = 42.05 years, *SD* = 16.49 years) were initially recruited for the study. The participants were recruited via advertisements in the mass mail systems of a university and on the WeChat social media platform. The participants were required to be (1) aged 18 or above, (2) mainland Chinese, and (3) living in mainland China during the study period. Data on 21 participants were excluded from the analysis due to technical problems encountered during the online experiment (e.g., the manipulation materials could not be properly played due to a poor Internet connection). Finally, data on 193 participants (70% female; age range: 18–81 years, *M* = 40.23 years, *SD* = 15.59 years) were included in the analysis (see Table 2). Based on power analysis conducted using the G*Power software package [42], the sample had statistical power of 80% to detect a medium effect size of 0.25 in an analysis of variance with two conditions.

### 5.2. Procedure

We built an online questionnaire on Qualtrics (https://www.qualtrics.com/ (accessed on 17 August 2021)). After signing a consent form, the participants were randomly assigned to the experimental condition or the control condition. In the experimental condition (COVID-19 condition), the participants were asked to watch a 3 min video showing street view during the COVID-19 lockdown in Wuhan (the city in which the first confirmed case of COVID-19 was reported), China, recorded in February 2020. The video displayed scenarios of the emergency rescue in hospitals, social distancing between family members, etc., all of which might make participants aware of vulnerability of human beings when facing infectious diseases. In the control condition, the participants were asked to watch a 3 min video of street view and interpersonal interactions in Hangzhou (a well-known city similar to Wuhan in China), recorded in 2018. The video clip in the control condition showed the daily life of healthy people, reunion dinners with friends and family, and a crowded tourist resort in the good weather. There was no information related to diseases in the video of Hangzhou. After watching the respective videos, the participants in the two conditions were asked to recall “a vivid personal experience relevant to the content of the video,” and write down the details of this experience by answering five questions (“When did it happen?”, “Where did it take place?”, “Who was involved?”, “What happened?”, and “What did you feel/think?”). To control for the accuracy of the recalled memories, the participants were asked to rate the accuracy level of their experiences from 1, very inaccurate, to 5, very accurate. Next, the participants responded to questions measuring their pro-environmental behavior, empathy, and demographic characteristics. The study was approved by the Human Ethics Review Committee of the Education University of Hong Kong. Each of the participants received a supermarket coupon of approximately US$7 in value after completing the 1 h study.

### 5.3. Measures

All of the measures were translated into Chinese and back-translated by independent translators, as recommended by Brislin [43].

#### 5.3.1. Pro-Environmental Behaviors

Pro-environmental behaviors were measured by an eight-item scale developed by Müller and Kals [44]. The participants indicated their agreement with eight items related to environmental protection on a 6-point Likert scale ranging from 1, *completely disagree*, to 6, *completely agree*. This measure assesses pro-environmental behaviors at the individual level, from general behaviors (e.g., “I do not wish to disturb or harm endangered animals or plants while in nature”) to specific behavior (e.g., “I am willing to contribute to the protection of the environment by reducing my car use (as a driver or passenger) and by switching to public transport or using a bicycle instead” (α = 0.84)).

#### 5.3.2. Empathy

The participants were asked to rate the intensity of their experience of empathy at a particular moment after watching the video on a 5-point Likert scale ranging from 1, *not at all*, to 5, very strong.

#### 5.3.3. Demographics

The participants’ age, gender (0 = *female*; 1 = *male*), education level (0 = *below college level*; 1 = *college level and above*), occupation status (0 = *did not have a job*; 1 = *had a job*), and socioeconomic status (SES) on a scale ranging from 1, *lowest*, to 10, *highest,* were recorded [45].

## 6. Study 2 Results

Controlling for age, gender, education level, job status, SES, and accuracy of memory, there was a significant main effect of condition (*b* = 0.27, *SE* = 0.11, *t* = 2.532, *p* = 0 .012, with 95% CI (0.06, 0.48)). The participants in the experimental condition reported a higher level of pro-environmental behavior commitment (*M* = 5.22, *SD* = 0.70) than their counterparts in the control condition did (*M* = 4.97, *SD* = 0.78). In addition, the participants in the experimental condition reported a higher level of empathy (*M* = 3.67, *SD* = 1.15) than did the participants in the control condition (*M* = 3.14, *SD* = 1.16) (*b* = 0.54, *SE* = 0.17, *t* = 3.184, *p* = 0.002, with 95% CI (0.20, 0.87)).

The mediation analysis using the PROCESS Macro version 3.5 [41] with condition as the independent variable, empathy as the mediator, and pro-environmental behavior as the dependent variable revealed a significant mediation effect. As shown in Figure 3, participants in experimental condition reported more empathy (*b* = 0.54, *SE* = 0.17, *p* = 0.002, 95% CI = (0.20, 0.87)) than did the control group. Empathy was associated with more pro-environmental behaviors (*b* = 0.12, *SE* = 0.05, *p* = 0.009, 95% CI = (0.03, 0.21)). After taking the effect of empathy into account, the direct effect size of the mediation was *b* = 0.23, *SE* = 0.11, *p* = 0.037, 95% CI = (0.01, 0.44)). Importantly, the indirect effect was 0.06 (95% CI = (0.01, 0.14)), showing that empathy was a significant mediator on increasing pro-environmental behaviors (Figure 3). The above results remained significant when not controlling for the covariates.

## 7. Study 2 Discussion

In Study 2, we manipulated infectious disease vulnerability and examined its impact on pro-environmental behaviors. We found that the participants in the experimental condition reported greater engagement in pro-environmental behaviors than did the participants in the control condition, and that this difference was mediated by empathy, such that greater infectious disease vulnerability induced more empathy and led to greater engagement in pro-environmental behaviors.

## 8. General Discussion

Based on the social identity model of pro-environmental action (SIMPEA) and using a combined methodology, namely, analysis of large-scale survey data from the WVS and an experiment, we found that infectious disease vulnerability increased pro-environmental behaviors at both the societal and the individual level. We found that at the societal level, the infectious disease vulnerability of a society increased the environmental concern of that society’s citizens, and therefore increased their pro-environmental behaviors. These findings are consistent with the social identity model of pro-environmental action.

At the individual level, we identified empathy as a mechanism that explains the relationship between infectious disease vulnerability and pro-environmental behaviors: infectious disease vulnerability induces empathy, which in turn encourages pro-environmental behaviors. As previous research has suggested, empathy is an affective state that promotes prosocial behaviors [30], such as pro-environmental behaviors [32].

However, other mechanisms may underlie the relationship between infectious disease vulnerability and pro-environmental behaviors. At the individual level, previous research has found that mortality salience increases pro-environmental behaviors [46,47]. This association is particularly strong for participants whose pre-environmental norms are salient [46] and for those who gain self-esteem from pro-environmental actions [44]. These findings may suggest that infectious disease vulnerability increases mortality salience, and therefore promotes environmental behaviors. They may also suggest that personal norms and values moderate the relationship between infectious disease vulnerability and pro-environmental behaviors. For example, the association may be weak among people who do not link the spread of the infectious disease with environmental disruption and those who do not prioritize pro-environmental goals. At the societal level, the association may be strengthened by appropriate policies, the establishment of pro-environmental organizations, and the provision of education regarding the relationship between infectious diseases and environmental problems. Future studies should investigate these alternative explanations.

We acknowledge that there were two limitations in the current study. First, we adopted data of WVS in Study 1 in which environmental concern was measured by hypothetical questions but not actual behaviors. Because of the nature of second-hand datasets, we relied on self-reported environmental concern. Future studies may improve the measure by using actual donation behaviors to test environmental concerns. Second, Study 1 was based on self-report data and Study 2 was based on an experiment. Future studies could validate the findings in a longitudinal design by comparing environmental concerns before and after infectious diseases. Despite these limitations, the studies reported here have several implications for the literature on pro-environmental behaviors. Most previous studies in this area have focused on how environmental problems, such as environmental disruption and air pollution, lead to the spread of infectious diseases [4,8]. Few studies have examined whether the spread of infectious diseases may in turn encourage people to pay more attention to protecting the environment and therefore engage in more pro-environmental behaviors. This information is important because it may identify a factor enhancing pro-environmental behaviors that has been neglected in previous studies. In addition, our findings suggest that perceived infectious disease vulnerability influences human psychological and behavioral activities. Although the concept of infectious disease vulnerability has been used in studies in the area of public health, few psychology studies have examined its implications. However, as demonstrated by the current studies, infectious disease vulnerability may influence human behaviors at both the individual and societal levels. In addition, our findings suggest that policymakers and practitioners could highlight individuals’ vulnerability to infectious diseases to more effectively encourage pro-environmental behaviors. Even more importantly, inducing empathy by enhancing knowledge of people’s suffering during outbreaks of infectious diseases may strengthen pro-environmental behaviors.

## 9. Conclusions

Using survey data covering 53 societies and an experiment, we found that infectious disease vulnerability increased pro-environmental behaviors at both the societal and individual levels. At the individual level, this association was mediated by empathy, such that greater infectious disease vulnerability increased empathy and therefore induced more pro-environmental behaviors. These findings shed light on the important role played by infectious disease vulnerability in promoting pro-environmental behaviors.

## Figures and Tables

**Figure 1 ijerph-18-08687-f001:**
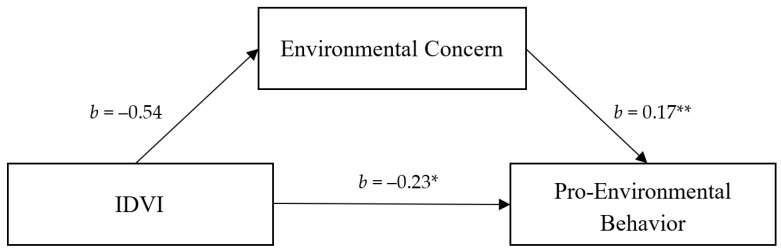
Note. Reported are standardized regression coefficients and standardized indirect effect with bootstrapped 95% confidence interval, ** *p* < 0.01, * *p* < 0.05.

**Figure 2 ijerph-18-08687-f002:**
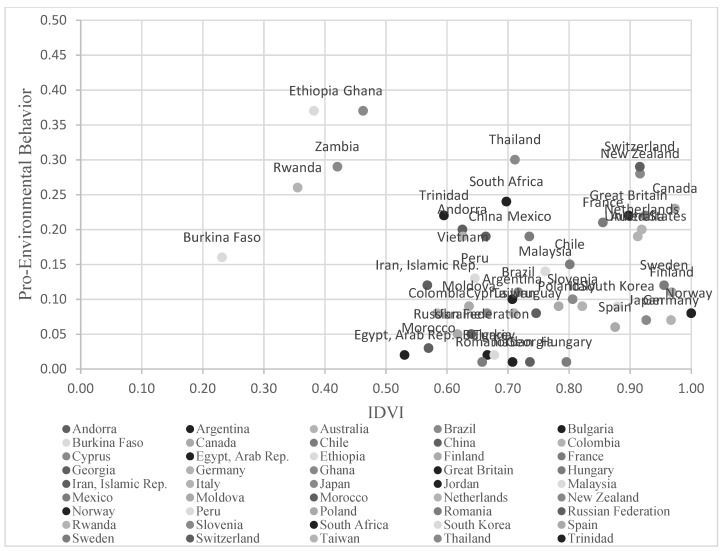
Correlation between Infectious Disease Vulnerability Index (IDVI) and pro-environmental Behavior in Study 1.

**Figure 3 ijerph-18-08687-f003:**
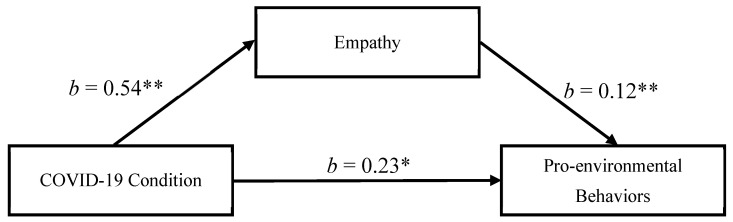
Empathy as mediator of the relationship between condition and pro-environmental behaviors in Study 2. Note. Reported are standardized regression coefficients and standardized indirect effect with bootstrapped 95% confidence interval, ** *p* < 0.01, * *p* < 0.05.

**Table 1 ijerph-18-08687-t001:** Descriptive information of the 56 societies in Study 1.

Society/Region	N	IDVI	GDPPC	Age	Female (%)	Pro-Environmental Behaviors			Environmental Concern		
				Mean (*SD*)		Mean (*SD*)	Median	Response Rate (%)	Mean (*SD*)	Median	Response Rate (%)
Andorra	1003	0.63	41,282.02	40.26 (13.68)	47.76	0.20 (0.52)	0.00	99.90	2.41 (0.62)	2.33	99.70
Argentina	1002	0.71	5109.85	42.55 (17.62)	52.79	0.10 (0.36)	0.00	100.00	2.15 (0.68)	2.33	97.51
Australia	1421	0.91	33,999.24	51.12 (16.51)	54.71	0.19 (0.49)	0.00	92.17	2.44 (0.68)	2.33	99.06
Brazil	1500	0.72	4790.44	39.98 (15.70)	52.40	0.11 (0.41)	0.00	98.67	2.23 (0.62)	2.33	99.80
Bulgaria	1001	0.67	3869.53	46.80 (16.58)	51.95	0.02 (0.16)	0.00	99.90	2.28 (0.72)	2.33	95.30
Canada	2164	0.97	36,266.19	46.96 (17.60)	52.06	0.23 (0.56)	0.00	99.26	2.58 (0.64)	2.67	99.31
Chile	1000	0.80	7598.53	46.69 (17.92)	52.50	0.15 (0.40)	0.00	100.00	2.18 (0.75)	2.33	97.90
China	1991	0.66	1753.42	40.09 (14.22)	48.97	0.19 (0.52)	0.00	99.10	2.80 (0.55)	3.00	92.06
Taiwan	1227	0.71	N/A	42.38 (16.05)	49.63	0.08 (0.34)	0.00	100.00	2.70 (0.56)	2.67	99.67
Colombia	3025	0.58	3404.19	36.96 (13.97)	50.02	0.08 (0.35)	0.00	100.00	N/A	N/A	0.00
Cyprus	1050	0.67	24,959.26	40.90 (15.69)	53.96	0.08 (0.31)	0.00	97.62	2.51 (0.65)	2.67	99.62
Ethiopia	1500	0.38	162.43	29.93 (10.21)	48.53	0.37 (0.63)	0.00	97.27	2.71 (0.68)	2.67	99.73
Finland	1014	0.97	39,040.29	47.02 (17.05)	52.07	0.11 (0.36)	0.00	99.41	2.51 (0.68)	2.67	99.41
France	1001	0.86	34,760.19	47.61 (18.17)	52.15	0.21 (0.54)	0.00	99.90	N/A	N/A	0.00
Georgia	1500	0.74	1642.76	45.41 (17.19)	52.93	0.01 (0.08)	0.00	99.67	2.42 (0.72)	2.33	94.27
Germany	2064	0.97	34,507.37	47.60 (16.96)	51.70	0.07 (0.30)	0.00	98.89	2.00 (0.74)	2.00	99.22
Ghana	1534	0.46	491.95	33.86 (14.07)	49.41	0.37 (0.66)	0.00	97.07	2.79 (0.66)	3.00	99.09
Guatemala	1000	0.48	2077.83	33.74 (12.52)	51.30	N/A	N/A	0	2.52 (0.63)	2.67	99.40
Hong Kong	1252	N/A	26,649.75	44.31 (15.81)	52.16	0.04 (0.29)	0.00	100.00	2.55 (0.45)	2.67	99.36
Hungary	1007	0.80	11,200.58	46.66 (17.31)	53.33	0.01 (0.13)	0.00	99.80	2.12 (0.68)	2.00	98.81
India *	2001	0.49	714.86	41.37 (14.71)	43.09	0.70 (0.65)	1.00	100.00	2.60 (0.65)	2.67	79.06
Indonesia *	2015	0.56	1263.29	36.10 (13.94)	47.74	0.57 (0.80)	0.00	97.07	2.33 (0.52)	2.33	97.87
Iran	2667	0.57	3246.05	32.69 (12.77)	49.89	0.12 (0.39)	0.00	99.14	2.51 (0.49)	2.67	99.81
Italy	1012	0.82	32,043.14	45.62 (15.62)	50.10	0.09 (0.33)	0.00	98.91	2.22 (0.59)	2.33	98.02
Japan	1096	0.93	37,217.65	48.15 (15.74)	55.93	0.07 (0.34)	0.00	95.53	2.48 (0.66)	2.67	94.34
Jordan	1200	0.71	2214.02	37.70 (14.25)	50.25	0.01 (0.13)	0.00	100.00	2.19 (0.66)	2.00	97.83
South Korea	1200	0.88	18,639.52	42.20 (14.18)	50.58	0.09 (0.34)	0.00	99.75	2.47 (0.51)	2.33	99.92
Malaysia	1201	0.76	5587.03	31.84 (11.93)	50.12	0.14 (0.43)	0.00	99.92	2.32 (0.55)	2.33	100.00
Mali *	1534	0.18	488.83	37.21 (14.77)	49.61	0.70 (0.86)	0.00	83.77	2.70 (0.62)	2.67	98.50
Mexico	1560	0.73	8277.67	39.69 (15.72)	50.83	0.19 (0.53)	0.00	98.08	2.65 (0.51)	2.67	99.62
Moldova	1046	0.64	1034.71	42.78 (16.85)	52.68	0.09 (0.35)	0.00	100.00	2.38 (0.66)	2.33	99.90
Morocco	1200	0.57	2018.03	37.10 (13.28)	50.67	0.03 (0.19)	0.00	96.50	1.99 (0.69)	2.00	98.83
Netherlands	1050	0.92	41,979.06	44.60 (17.48)	51.14	0.20 (0.49)	0.00	97.43	N/A	N/A	0.00
New Zealand	954	0.92	27,751.07	49.25 (16.39)	54.98	0.28 (0.59)	0.00	87.00	2.30 (0.63)	2.33	95.39
Norway	1025	1.00	66,810.48	45.78 (16.06)	49.85	0.08 (0.32)	0.00	100.00	2.71 (0.86)	3.00	99.41
Peru	1500	0.65	2729.50	37.61 (14.88)	51.00	0.13 (0.47)	0.00	99.73	2.55 (0.51)	2.67	97.73
Poland	1000	0.78	8021.00	45.21 (17.84)	52.40	0.09 (0.34)	0.00	99.80	2.19 (0.74)	2.33	97.80
Romania	1776	0.66	4617.93	48.68 (17.38)	54.45	0.01 (0.11)	0.00	99.83	2.00 (0.74)	2.00	94.43
Russia	2033	0.64	5323.47	42.84 (17.08)	54.50	0.05 (0.24)	0.00	99.16	N/A	N/A	0.00
Rwanda	1507	0.36	292.00	34.65 (14.14)	50.63	0.26 (0.58)	0.00	94.69	2.41 (0.61)	2.33	99.20
Vietnam	1495	0.63	687.48	40.75 (15.85)	48.70	0.19 (0.56)	0.00	100.00	3.01 (0.50)	3.00	96.25
Slovenia	1037	0.81	18,098.91	46.19 (17.84)	53.52	0.10 (0.38)	0.00	100.00	2.50 (0.66)	2.67	97.88
South Africa	2988	0.70	5383.66	37.26 (16.54)	49.97	0.24 (0.51)	0.00	100.00	2.27 (0.71)	2.33	95.82
Spain	1200	0.88	26,419.30	46.05 (18.50)	51.50	0.06 (0.28)	0.00	99.92	2.06 (0.64)	2.00	98.17
Sweden	1003	0.96	43,164.00	47.80 (17.03)	49.65	0.12 (0.35)	0.00	99.40	2.77 (0.57)	3.00	99.70
Switzerland	1241	0.92	54,952.68	52.54 (16.36)	54.31	0.29 (0.56)	0.00	99.11	2.59 (0.69)	2.67	99.76
Thailand	1534	0.71	2894.06	45.35 (15.74)	50.98	0.30 (0.64)	0.00	98.83	2.67 (0.42)	2.67	99.93
Trinidad and Tobago	1002	0.59	12,327.23	40.57 (16.36)	50.00	0.22 (0.53)	0.00	100.00	2.42 (0.59)	2.33	100.00
Turkey	1346	0.68	7384.25	36.49 (13.85)	49.85	0.02 (0.19)	0.00	100.00	2.64 (0.58)	2.67	97.77
Ukraine	1000	0.62	1826.93	45.63 (17.40)	55.00	0.05 (0.26)	0.00	99.30	2.06 (0.67)	2.00	96.70
Egypt	3051	0.53	1187.52	41.71 (14.61)	48.97	0.02 (0.16)	0.00	100.00	1.99 (0.69)	2.00	99.84
United Kingdom	1041	0.90	42,030.29	45.70 (18.51)	51.97	0.23 (0.54)	0.00	98.56	N/A	N/A	0.00
United States	1249	0.92	44,114.75	45.90 (16.89)	51.72	0.22 (0.54)	0.00	99.12	2.35 (0.65)	2.33	97.52
Burkina Faso	1534	0.23	407.00	34.27 (13.84)	48.88	0.16 (0.48)	0.00	91.33	2.66 (0.65)	2.67	94.20
Uruguay	1000	0.75	5226.94	46.53 (18.65)	55.60	0.08 (0.33)	0.00	99.40	2.15 (0.58)	2.00	97.20
Zambia	1500	0.42	702.74	29.79 (11.88)	49.33	0.29 (0.58)	0.00	94.80	2.29 (0.70)	2.33	94.93

Note. IDVI = Infectious Disease Vulnerability Index. GDPPC = per capita gross domestic product. * The scores of India, Indonesia, and Mali were detected as outliers, and were exclude in data analysis.

**Table 2 ijerph-18-08687-t002:** Descriptive information of variables used in Study 2.

Variables	Experimental Condition	Control Condition	*t*-Test	*p*
	(*N* = 86)	(*N* = 107)		
	Mean (*SD*)	Mean (*SD*)		
Age	40.19 (14.70)	40.27 (16.33)	0.04	0.97
Accuracy of memory	4.17 (1.07)	4.30 (0.72)	0.93	0.354
Socioeconomic status	2.91 (1.60)	2.75 (1.35)	−0.74	0.462
Empathy	3.67 (1.15)	3.14 (1.16)	−3.16	0.002
Pro-environmental behaviors	5.22 (0.70)	4.97 (0.78)	−2.29	0.023
Female percentage	68.60%	71.00%	0.36	0.718
Education level (college level and above)	70.90%	74.80%	0.59	0.555
Job status (has a full-time job)	47.70%	43.90%	−0.52	0.606

## Data Availability

The data presented in this study are available on request from the corresponding author. The data are not publicly available due to privacy.

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
