# Peer review of "Learning from COVID-19: Infectious Disease Vulnerability Promotes Pro-Environmental Behaviors"

_ijerph, 2021, doi:10.3390/ijerph18168687_

Round 1

Reviewer 1 Report

I appreciated the opportunity to read your paper. I enjoyed it and found your paper extremely well written. Your research questions are clear and your data appeared to support your conclusions. The subject matter is of high interest especially as the planet is suffering great environmental degradation.

There are two small English language goofs that I saw.

Page 2 line 69 This argued, however, has not been empirically tested yet. SHOULD BE This argument, however

Page 3 line 111 Using experimental designs, Berenguer (2007) found that inducing empathy increased pro-environmental attitudes and behaviors. SHOULD BE Using experimental design, Berenguer

Author Response

Dear Reviewer 1

We would like to thank you for your positive comments. We feel grateful that you think our topic is of high interest and our paper is extremely well written. Thank you so much for all your encouragement. 

Thanks for pointing out two types and errors in the manuscript. We have addressed the errors you identified. Thanks again for all your support!

Best Regards,
Da Jiang
Department of Special Education and Counselling,
The Education University of Hong Kong  

Reviewer 2 Report

Thank you for the opportunity to review this paper. This topic is very important and presented properly could impact motivation and understanding of human behavior and response to intricately connected issues that this planet faces. This study used two designs, ecological and experimental to assess how disease susceptibility (specifically, vulnerability to viruses) impacts behaviors that are positive for the environment. Secondarily, they sought to determine if this relationship was modified by environmental concern. They found a significant association between vulnerability and pro-environmental behaviors that was also modified by environmental concern across both ecological and individual studies.

Specific comments:

Line 26-27: Consider updating the global statistic on COVID-19 with something more recent than nearly a year ago.

Line 69: argument instead of “argued”

Line 124: Please add the abbreviation used later to the first mention of the World Value Survey

Line 174-175: If you are going to list 2 of the 3 examples, please just list the third one as well for transparency and clarity.

Line 192-195: The sentence reads that these are positive associations but the betas and 95% CIs are all negative.

Figure 1: Based on this figure there are some significant outliers (Mali, India, Indonesia) that could really be pulling/driving the association between the two variables. Although they are real instances, I wonder if you did a sensitivity analysis without their data and if the relationship would hold. This is an ecological analysis and thus highly prone to ecological fallacy. Without the sensitivity analysis, it’s hard to believe that the results hold at the societal level across multiple societies.

Table 1: Given that Pro-environmental Behaviors and Environmental Concern were collected on a discrete scale and not a continuous one, it would be more appropriate to present how many responses were received for each discrete answer by country and a median than a mean. Although, for me, the ideal presentation would be all three since they each tell different aspects of the story. Additionally, it could be more informative to use the discrete form of each variable in the model rather than the mean.

Table 2: Did you do any bivariate statistical tests between the experimental arm and the control arm for the demographics and variables to let the reader know if the differences between the groups reached statistical significance?

Line 245-247: How was the randomization done? Please add details and discuss the obvious bias that exists across the experimental results as shown in Table 2 by the differences in mean across different relevant variables.

Line 249: in not “win”

Line 277-279: Can you please expand on this measure of empathy? Was it related to the experience that they wrote about after watching the 3 min thing or was it directly related to the 3 min clip? Or was it just used as a tool to collect memory accuracy?

Line 303-304: It’s a little unclear how having an experimental and non-experimental arm and playing a video is manipulating the infectious disease vulnerability.

General Comments:

This paper is well written with sufficient background and presentation of the problem investigated. The presentation of the results should be improved to avoid any potential confusion by readers that do not read carefully. Negative and positive associations are sometimes hard for people to understand, particularly when the scales of the variables being related are not very familiar.

Lower score = greater environmental concern – this is very counter-intuitive particularly because of the use of “pro” in front of environmental activity. Plus coupled with the use of negative and positive association terminology the results can be easily misread or misinterpreted, particularly by those that do not give it and the instrument section a careful read.

This paper would benefit from a deeper discussion of the questions used to measure/quantify pro-environmental behavior and environmental concerns. Specifically, their strengths and limitations should be discussed with the understanding that they are proxies to the true measurement desired.

Possible confounders in the model/methods (please discuss the confounders in the discussion and how they may impact the relationship):

  • I know that you adjusted for GDPPC, but after adjustment how is the activeness as measured by the instrument in low-income/low-resource countries compared to high-income/high-resource countries. What does it look like per capita…  are there residual biases and confounding that isn’t accounted for with the adjustment for GDPPC because you would expect IDVI and this low score good pro-environmental activity metric to be negatively associated just due to resource constraints both on the person level and the societal level.
  • Most of the societal level averages for Environmental Concern are right in the middle of the scale with not very much variation when considered as a mean. The other factor that this lack of diversity in mean response makes me think about political inactivity. By asking someone if they would contribute money or pay more taxes, you are by proxy asking them if they would give $$ to their government to help the environment. What was the third question to evaluate environmental concern? Was there a question that just asked people how important they thought the environment was, maybe even related to other things that people generally think are important that could be a better (or alternative) metric? Additionally, I would love to see the mean responses by sub-question of a given instrument, it’s a bit alarming to see how low the mean responses are.
  • The modeling as far as I understand from the paper doesn’t take into consideration the abundance of data that you have. My understanding is that the means of each variable for each country were used in the models. This negates all of the variation and potentially interesting findings that could be seen by using all of the data. For instance, if repeated measures within-country were used, you could use the country as a random variable (slope, intercept, or both). I understand that PROCESS cannot do this, but MLMED can. Might be worth considering.
  • The experimental arm was a bit bias toward, higher SES, higher empathy, higher pro-environmental behaviors, and more with a full-time job. Can you present the statistics for the variables across experimental arms?

Author Response

Dear Reviewer 2
We would like to thank you for your constructive comments, which have greatly helped me to improve the paper. As you instructed, I have marked the changes using the “Track Changes” function in the manuscript. You will find the point by point responses to all suggestions of yours in the attached response letter. 

Thank you very much again for all the helpful comments. 

Best Regards,
Da Jiang
Department of Special Education and Counselling,
The Education University of Hong Kong  
